# Evaluating the Role of Corrals and Insects in the Transmission of Porcine Cysticercosis: A Cohort Study

**DOI:** 10.3390/pathogens12040597

**Published:** 2023-04-14

**Authors:** Eloy Gonzales-Gustavson, Ian W. Pray, Ricardo Gamboa, Claudio Muro, Percy Vilchez, Luis Gomez-Puerta, Ana Vargas-Calla, Gabrielle Bonnet, Francesco Pizzitutti, Hector H. Garcia, Armando E. Gonzalez, Seth E. O’Neal

**Affiliations:** 1Departamento de Salud Animal y Salud Pública, Universidad Nacional Mayor de San Marcos, Lima 15081, Peru; 2School of Public Health, Oregon Health & Science University and Portland State University, Portland, OR 97239, USA; 3Center of Global Health, Universidad Peruana Cayetano Heredia, Lima 15202, Peru; 4Centre for the Mathematical Modeling of Infectious Diseases, London School of Hygiene and Tropical Medicine, London WC1H 9SH, UK; 5Geography Institute, Universidad San Francisco de Quito, Quito 170157, Ecuador; 6Cysticercosis Unit, National Institute of Neurological Sciences, Lima 15003, Peru

**Keywords:** porcine cysticercosis, cohort, free roaming, corrals, insects, seroincidence

## Abstract

The widespread dispersion of pigs infected with cysticercosis across endemic villages, low cyst burden among infected pigs, and low prevalence of taeniasis all suggest that pig ingestion of human feces is not the only mode of transmission for *Taenia solium*. Our objective was to evaluate the risk of porcine cysticercosis associated with exposure to human feces, dung beetles, and flies in an endemic community setting. We used a cluster-randomized cohort design to compare the risk of developing antibodies and infection among 120 piglets raised in either free-roaming (FR), standard corral (SC), or netted corral environments (NC). We collected monthly blood samples to detect serum antibodies and necropsied all pigs after 10 months to identify cysts. A total of 66 piglets developed antibodies with the relative risk of seropositivity in FR vs. all corralled pigs increasing significantly after 18 weeks. Of 108 necropsied pigs, 15 had *T. solium* cysts, all belonging to the FR group. Corrals were protective against infection but less so against seropositivity. NC, which did not completely exclude insects, did not provide added protection against seropositivity as compared to SC. The results of this study suggest that dung beetles and flies do not play an important role in infection.

## 1. Introduction

*Taenia solium* cysticercosis is a leading cause of acquired epilepsy across Latin America, Asia, and Africa, and is among the most common parasitic causes of neurologic disease worldwide. In regions where the disease is endemic, 30% or more of seizure disorders are attributed to brain infection by *T. solium* larval cysts [1]. The disease is also of increasing importance in the United States where total hospitalization charges for cysticercosis amount to over $100 million per year [2]. Although the International Task Force for Disease Eradication has included cysticercosis on its short list of potentially eradicable diseases for over two decades [3], there are still significant knowledge gaps regarding how *T. solium* is transmitted between human and pig hosts at the community level. These gaps (described below) limit the ability to design effective and sustainable strategies for control, regional elimination and ultimately eradication [4].

Among the most puzzling aspects of transmission yet to be explained is how *T. solium* eggs attain widespread dispersal within affected communities. Humans are the sole definitive host of the egg-producing intestinal tapeworm. Pigs are the intermediate host and become infected with tapeworm larvae upon ingestion of tapeworm eggs. While pig ingestion of *T. solium* eggs in human fecal depositions has long been considered the only method of transmission to pigs, this appears inconsistent with the results of epidemiologic studies from around the globe, as these show very widespread exposure and infection of pigs within endemic communities [1,5,6,7,8]. This pattern cannot be explained by direct fecal ingestion alone in a context in which the lifespan of pigs raised for consumption is short (typically less than one year), human intestinal tapeworm infection prevalence is low (typically less than 3%), and tapeworm lifespan is relatively long (estimated at approximately 2 years) [9,10]. Hence, mechanisms of egg dispersal away from human fecal depositions are likely to play an important role in transmission. This hypothesis is further supported by the rapidity with which *T. solium* repopulates communities that have undergone mass chemotherapy, which could be explained by a persistent environmental reservoir of infectious eggs [4]. More complete understanding of potential alternate transmission mechanisms is needed if we are to reach the desired goal of preventing epilepsy and other neurologic morbidity caused by this parasite.

Insects are known to play an important role in the transmission of many cestode parasites including those of the Taenidae family. Dung beetles (Scarabaeidae family) have the potential to widely disperse parasite eggs given their behavior of rolling and pushing dung balls to create brooding nests. This rolling behavior is a well-described mechanism for plant seed dispersal and could contribute to parasite egg dispersal as well. Dung beetles have been shown to act as mechanical vectors of infectious eggs for other *Taenia* species such as *T. saginata* and *T. hydatigena* that are typically co-endemic with *T. solium* [11]. In a series of preliminary studies, we have demonstrated that dung beetles can harbor *T. solium* eggs in their intestinal tract and that eggs can remain viable there for a period of several weeks [12,13]. We have also shown that experimental infection of pigs with dung beetles carrying *T. solium* eggs is an efficient route of transmission that results in cysticercosis [14]. While these results suggest that dung beetles could indeed play a role in the dispersion of *T. solium* eggs, the importance of this potential transmission pathway in endemic settings is unclear. We conducted a controlled field experiment in northern Peru to evaluate the extent to which dung beetles and flies are involved in *T. solium* transmission in a typical endemic transmission setting.

## 2. Materials and Methods

### 2.1. Study Design and Outcomes

We used a cluster-randomized cohort study to compare the risk of developing antibodies and infection among 120 seronegative piglets raised in three distinct risk environments: free-roaming (FR, expected to be exposed to human feces and flying insects), standard corrals (SC, expected to be protected from human feces, exposed to flying insects), or netted corrals (NC, expected to be protected from human feces and flying insects). Piglets were randomly assigned and introduced into all three environments in four separate rounds (months 0, 3, 6, and 9) of 30 piglets each in order to control for seasonal effects. All cohort pigs underwent monthly blood sampling and were followed for a period of 10 months, at which point they were humanely euthanized and dissected. The primary outcome was the cumulative incidence of developing serum antibodies against cysticercosis in each group of pigs using lentil lectin-bound glycoprotein enzyme-linked immunoelectrotransfer blot assay (LLGP-EITB). The secondary outcome was the proportion of pigs in each group found to be infected with cysts upon necropsy. Traps for dung beetles and flies were placed in all environments to monitor exposure to flying insects; captured insects were evaluated for the presence of *Taenia* sp. eggs (dung beetles) or DNA (flies).

### 2.2. Study Setting

The study occurred in two adjacent rural villages from the Matapalo District in the Tumbes Region of northern Peru where *T. solium* is endemic (taeniasis prevalence 1.4–2.7%; unpublished). This tropical coastal region experiences a hot rainy season from February through May (average high temperature 32 °C; ~10 cm precipitation) and is slightly cooler and much drier for the remainder of the year (average high temperature 30 °C; <1 cm precipitation). Subsistence agriculture is the main economic activity in this impoverished region and pigs are raised using the scavenging method in which they are allowed to roam and forage. Human outdoor defecation is common and insects, including dung beetles, are readily visible. There were no ongoing control programs for *T. solium* in these communities, which comprise approximately 1800 people and 500 pigs.

### 2.3. Identification of Hotspots and Selection of Households

We used GPS receivers and post-processing for sub-meter accuracy (Trimble, Sunnyvale, CA, USA) to map all houses in the study communities on the ArcGIS platform (ESRI, Redlands, CA, USA). We then conducted 4 separate serosurveys (at months 0, 3, 6, 9) of all pigs in these villages in order to identify transmission hotspots at each timepoint. We went door-to-door to capture all pigs and collect a venous blood sample from the anterior vena cava. Sera samples were processed for the presence of antibodies against cysticercosis using LLGP-EITB. We then used Getis Ord local Gi* statistic to identify transmission hotspots based on the number of pigs with strong reaction on LLGP-EITB (4 or more reactive bands present), and these were ranked based on their *p*-value. We selected three households within distinct hotspots for placement of seronegative cohort piglets at each timepoint (12 households total). Criteria used to select households included *p*-value of the hotspot, number of positive pigs in the household, and distance from previously selected households.

### 2.4. Construction of Corrals and Animal Maintenance

We prepared three experimental exposure environments at each selected household for placement and maintenance of cohort pigs (Figure 1). For the SC group, we constructed 15 m^2^ open-air corrals that restricted pig movement (no access to human feces) but allowed contact with soil and insects. For the NC group, we constructed similar 15 m^2^ corrals but enclosed these in a double layer of polypropylene mesh; these corrals served to restrict pig movement and limit exposure to flying insects. Overlapping metal sheets were also buried 50 cm deep in the ground under the NC walls to exclude tunneling dung beetles. For the FR group, pigs were allowed to join the household herd and roam the village freely to scavenge for food, allowing exposure to both human feces and insects. Cohort pigs were maintained exclusively in their assigned environments for a period of 10 months. Our veterinary staff visited each participating household twice per day to feed and monitor the well-being of the pigs. Pigs in the SC and NC groups only were provided commercial pig feed and water from the household source.

### 2.5. Selection and Assignment of Animals

We sourced 10-week-old piglets in blocks of 30 from local granges in Tumbes, representing mixed breeds common to the region. To ensure that piglets were seronegative, we first screened the sow for serum antibodies (LLGP-EITB) or antigens (B158-B160 monoclonal antibody-based ELISA) to identify litters from disease-free mothers, then verified the seronegative status of all piglets using the same tests prior to introduction of the piglets into the study. We then randomly distributed the 30 piglets from each block to new hotspot households and their FR, SC, and NC exposure environments (4, 3, and 3, piglets in each environment, respectively, per household). In total, 120 seronegative piglets were included in the cohort and assigned to FR (*n* = 48), SC (*n* = 36), and NC (*n* = 36) exposure groups at 12 households; more piglets were assigned to FR due to the greater risk of premature loss or death of the free-roaming animals. Piglets were placed in December 2015, April 2016, June 2016, and October 2016.

### 2.6. Procedures for Measuring Outcomes

We collected a venous blood sample from the anterior vena cava of all cohort pigs each month, keeping the pigs inside their corrals to avoid the possibility of outside exposure or infection. All blood samples collected in the study were maintained temporarily on ice in coolers while teams were in the field, centrifuged daily in a field laboratory, frozen at −20 °C, and shipped to the Cysticercosis Unit of the National Institute of Neurosciences, Lima, for processing. Sera samples from the candidate piglets, village pigs during serosurveys, and cohort pigs were processed using the LLGP-EITB for presence of antibodies against cysticercosis using methods described elsewhere [15,16]. Briefly, the LLGP-EITB assay uses an enriched fraction of homogenized *T. solium* cysts containing seven *T. solium* glycoprotein antigens, GP50, GP42, GP24, GP21, GP18, GP14, GP13. Reaction to any of these seven glycoprotein antigens was considered positive and interpreted as evidence of probable exposure to *T. solium* eggs. However, the EITB-LLGP is now known to cross react on the GP50 band with *T. hydatigena* [17]. Because antibody reaction to the different families of glycoprotein antigens tend to develop, and regress, sequentially consistent with the life stage of the parasite, we also report the maximum number of reactive bands present at any time during the observation period. Sera from the candidate piglets were also processed for the presence of *Taenia* sp. antigens using Ag-ELISA and mAb set B158-B60 [18]. An optical density ratio > 1 was considered positive.

After 10 months of maintenance in their assigned exposure environments, cohort pigs were anesthetized using intramuscular ketamine (20 mg/kg) and xylazine (2 mg/kg), then euthanized by intravenous sodium pentobarbital (100 mg/kg). We conducted a detailed necropsy by systematically dissecting the entire carcass, and identifying any viable or degenerating cysts present in the brain, heart, liver, tongue and all skeletal muscles. Fine slices of less than 0.5 cm were used to dissect all inspected tissues.

### 2.7. Insect Collection

We placed insect traps in all exposure environments once per month to collect both dung beetles and flies. Twelve traps were placed at each selected household, four each for netted and standard corrals (one trap in each corner of the corral) and four placed approximately 10 m apart outside of the corrals. The traps consisted of an iron rebar cage to exclude animals. Inside 2 of the cages, a bait (gauze bag containing 150 g of 1:1 mixed pig feces and human feces from individual without taeniasis infection, as determined by negative ELISA coproantigen test and microscopy) was suspended 15 cm directly over a plastic cup with inverted funnel buried to the lip in dirt. Flying or crawling beetles attracted to the bait that fell into the trap were unable to escape. In the other 2 cages, a bait (150 g of pig feces without gauze) was placed directly on the soil without a pitfall trap; the bait itself and 10 cm of soil beneath it were collected and examined for the presence of dung beetles. On top of the rebar cages within corrals, we secured a plastic open bowl containing a fly attractant with insecticide (Agita 10 WG, Elanco, Greenfield, IN, USA). On top of the rebar cages outside of the corrals, we secured a paper plate covered with adhesive to trap flies that landed there. The Agita insecticide method was used only inside netted corrals to avoid unintentional intoxication of birds. In all cases, traps were placed so that they could not be accessed by study pigs or other animals. Traps were left in place for 48 h, after which time all beetles and flies were removed and counted, then stored separately in 2.5% potassium dichromate. In total, 1296 traps were placed during the study.

### 2.8. Microscopic Diagnosis

Identification of Taeniidae eggs in beetle intestines was carried out using the protocol described by Gomez-Puerta et al. [12] with some modifications. Briefly, beetles were dissected under a stereomicroscope and their intestines were placed on slides, a clearing solution (ethanol-phenol, 1:2 *v*/*w*) was added, and they were examined under an optical microscope under at 100× or 400× magnification.

### 2.9. Molecular Diagnosis

Flies were pooled (3 to 5 flies per pool) for molecular diagnosis of *T. solium*. DNA isolation was performed using a commercial kit (FastDNA Spin Kit for Soil, MP Biomedicals, Carlsbad, CA, USA) according to the manufacturer’s instructions. A PCR protocol to partially amplify the *T. solium Tsol18* protein gene was used for diagnosis, using a protocol and primers forward 5′-GGT TTG CTC TCA TCT TCT TGG TGG C-3′ and reverse 5′-CGA AGA TTT ATT CGT TAA CAT GAA AGG TC-3′ proposed by Gauci et al. [19].

### 2.10. Data Analysis

We used Cox regression analyses to evaluate the determinants of *T. solium* exposure [20]. The dependent variable was the time it took for a pig to become positive on any EITB band, in weeks. Seven explanatory variables were tested: exposure environment (each of the three cohorts corresponding to a different environment), placement round, sex, village, presence of other cestodes on necropsy (such as *T. hydatigena* and/or *E. granulosus* larval forms), pig age and weight at necropsy. To select variables for inclusion in the Cox regressions, we first assessed associations individually between each explanatory variable and the outcome using log-rank tests (threshold *p* < 0.1) (Table 1), and visually using Kaplan–Meier survival curves (Appendix A).

Before proceeding with Cox regression, we evaluated the data for conformity to the proportional hazards assumption by plotting log (−log) curves of the Kaplan–Meier survival function (Appendix A), and by performing a goodness of fit test (Appendix A). The explanatory variable ‘exposure environment’ was found to be a powerful predictor of LLGP-EITB result, but it did not satisfy the proportional hazards assumption (Appendix A). Consequently, we developed first a stratified (by exposure environment) Cox regression to analyze the effect of each variable independent of the exposure environment. We then proceeded with an extension of the Cox proportional hazard model for time-dependent variables to assess the effect of exposure environment along with other factors. In both cases, we applied stepwise elimination of variables. All explanatory variables used in the final model were also tested for conformity to the proportional hazard assumption. None of these variables failed the test using the *p*(PH) statistic.

The development of the extended Cox Regression for time-dependent variables was also informed by the evolution of the effect of the variable ‘exposure environment’ on the outcome (positive to LLGP-EITB) over time. Although the proportional hazard assumption was violated, there was an interval of time at the beginning of the experiment during which the log (−log) curves for the three exposure environments were parallel. Over this interval, the test of the proportional hazard assumption for the variable ‘exposure environment’ was not significant, and the extended Cox Regression for time-dependent variables was developed taking that interval into account.

In a final step, and based on the outcomes of the analyses described above, the explanatory variables ‘exposure environment’ and ‘rounds’ were converted into two-level categorical variables; for ‘exposure environment’, SC and NC were combined as they led to very similar results, and for ‘rounds’, the first round was combined with the fourth and the second round with the third (also because of similarity in results). All survival analyses were performed using R software (version 4.2.2) [21] with the package ‘survival’ [22].

### 2.11. Ethical Review

This study was reviewed and approved by the Institutional Animal Care and Use Committees at Oregon Health & Sciences University, Portland, Oregon, USA, and at the Universidad Peruana Cayetano Heredia, Lima, Peru. Treatment of animals adhered to the Council for International Organizations of Medical Sciences (CIOMS) International Guiding Principles for Biomedical Research Involving Animals.

## 3. Results

### 3.1. Seroincidence, Necropsy and Infection Risk

Of the 120 piglets included in the study, 36 were housed in NC, 36 in SC, and 48 in FR environments. A total of 66 piglets seroconverted over the study period with a seroincidence rate and 95% exact Poisson confidence interval of 0.02 (0.01–0.03), 0.02 (0.01–0.03) and 0.04 (0.02–0.05) cases per pig-week in NC, SC, and FR environments, respectively. Appendix A details the maximum number of reactive bands present throughout the entire observation period for pigs in each of exposure environments Necropsies performed on 108 pigs identified 15 pigs with cysts (median 4 cysts per infected pig, range 1–2387), including 10 with viable cysts. The 12 pigs that did not undergo necropsy included 11 pigs from FR and 1 from NC environments, and in all cases either the pig was lost, or the pig died and the condition of the carcass when discovered did not allow necropsy. All pigs with cysts at necropsy belonged to the FR exposure environment, with an incidence rate of 0.02 (0.01–0.03) cases per pig-week, assuming that those pigs were infected at the time they seroconverted and excluding the 11 pigs that were not necropsied. There were significant differences in the proportion of pigs with any cysts on necropsy between rounds (Appendix A) but not between sexes (Appendix A).

### 3.2. Seroconversion Hazard and Evolution over Time

The median time pigs were observed was 23 weeks (range 4–44 weeks). Figure 2 shows the duration during which each pig was observed. Table 1 describes the number of pigs, LLGP-EITB positive pigs, and Log rank test for sex, exposure environment, and round, the three variables for which there were significant differences in the preliminary analysis. No difference was found between villages, presence vs. absence of other cestodes, ages, or weight at necropsy (data not shown). The KM survival curve (Figure 3) revealed a sharp increase in the hazard for pigs in the FR environment at 18 weeks, while the hazard for pigs in the NC and SC environments remained relatively constant throughout the observed period. Additional KM survival plots comparing rounds, sex and other categorizations are shown in Appendix A.

The extension of the Cox proportional hazard model for time dependent variables analysis confirmed that hazards for FR and combined corral were similar during the first 18 weeks of evaluation (Wald test, *p* = 0.7); after that period, seropositivity hazard was 5.7 (95%CI 2.9–11) times higher in the FR group (Wald test *p*-value < 0.001) (Table 2). Differences in hazards between rounds and sex remained significant in the extended Cox model analyses as well.

### 3.3. T. solium Eggs in Insects

Traps were placed and inspected 1296 times during the study period. A total of 1424 beetles were collected, mainly of the Scarabidae family; all were dissected with the intestinal tracts examined using microscopy and none were found to be harboring *T. solium* eggs. We also collected 2948 flies, primarily *Musca domestica*, dividing these into pools of 3 to 5 flies; none of the pools were positive for *T. solium* DNA by PCR. There were more flies caught in the NC (1707) and FR (758) environments, than the SC (483). The unexpected higher number in the NC was due to an initial peak that occurred in the first months of the study only (Appendix A). In contrast, more beetles were found in FR (1044) environments throughout the study, with less in the NC (175) and SC (205) (Appendix A).

Excluding the early peaks for flies (excluding first 5 months, includes trap inspection 1152 times) and beetles (excluding first 6 months, includes trap inspection 1044 times) showed significantly more flies and beetles in FR (513 and 504, respectively) compared with NC (275 and 171, respectively) and SC (364 and 132, respectively) (*p* < 0.0001 for both) (Appendix A).

### 3.4. Pig Infection with Other Cestodes

We found pigs infected with other (non-*T. solium*) cestode parasites during necropsy, including larval forms of *T. hydatigena* and *Echinococcus granulosus;* 5 out of 35 (14%), 2 out of 36 (6%) and 14 out of 37 (38%) pigs in NC, SC, and FR environments, respectively, harbored other cestodes, with significantly higher infection rates in FR (Chi-squared test, *p* = 0.001). Finally, 9 of the 15 (60%) pigs with *T. solium* cysts at necropsy and 12 of the remaining 93 (13%) without it, harbored other cestodes at necropsy.

## 4. Discussion

The objective of this study was to evaluate the extent to which dung beetles and flies play a role in transmission of *T. solium* to pigs. We found no evidence to suggest that either insect plays a major role in the transmission of *T. solium* to pigs. On the contrary, the absence of pig cysticercosis infection in the standard corral environment, along with the complete absence of *T. solium* eggs in dissected beetles and of *T. solium* DNA in flies, suggest that other factors (e.g., alternate routes of low-burden egg exposure, variability in egg infectivity, variability in pig susceptibility) are more likely to explain dispersion and low-cyst burdens of infected pigs.

Despite our best efforts, we were unable to completely exclude dung beetles and flies from the netted corrals. We suspect, but cannot confirm, that these insect populations resulted from larvae present in the soil when the corral was constructed, and that the nets prevented the newly emerged insects from dispersing. After discovering this problem, we instigated periodic insecticide spraying within all netted corrals, which controlled but did not eliminate the problem. After the new measures, there were always more beetles and flies in the free roaming setting. If these were indeed trapped insect populations, the beetles and flies would not have been exposed to *T. solium* eggs from open human defecation outside the netted corrals, and hence would not be expected to contribute to transmission. The transient peaks of insects in the netted corrals did not appear to influence study outcomes, as they occurred in the first round in which there was the least seroconversion and no cases of pig infection. Because there were no differences in serologic or necropsy outcomes between the two corral environments, we combined all corralled pigs into one group for our final analysis.

The use of corrals to restrict pig access to human fecal depositions proved sufficient to completely prevent infection with *T. solium*. None of the 71 corralled pigs were found to have cysticercosis on necropsy, compared to 15 of 37 (41%) of free-roaming pigs. Porcine cysticercosis was somewhat more likely among pig cohorts that predominantly spanned the dry season, but there were otherwise no important risk factors that we could identify. The incidence of 0.02 infections per week should also be interpreted with caution as we placed pigs in areas with suspected high transmission within villages, so the measured rate is likely to overestimate that of the general population of pigs in the village. While strict corralling may seem an intuitive solution to control transmission, feasibility and acceptability of this approach would need to be carefully evaluated. Many households in endemic villages in this region have corrals, but these are used only to fatten pigs in the few weeks prior to sale. Pigs are allowed to roam and scavenge to avoid the economic costs and labor associated with feeding and watering corralled pigs.

It was interesting that corralling did not, however, completely prevent infection with other cestode species. *T. hydatigena* and *E. granulosis* were found in corralled pigs, albeit less frequently than in the free-roaming pigs (9.9% vs. 37.8%, respectively). *T. hydatigena* eggs in particular are highly capable of dispersing and causing infection in the intermediate hosts. In one observational study, Soay Sheep were found to be infected with *T. hydatigena* on St. Kilda Island in the absence of definite hosts, and were presumed to have been exposed by eggs transported in the feces of seagulls [23,24]. Later experiments showed that lambs grazing up to 80 m away from kenneled dogs infected with *T. hydatigena* tapeworms also developed infection with *T. hydatigena* metacestode cysts [5,25]. The source of the other cestode infections in our study is not clear. While it is possible that corralled pigs were exposed to the cestodes while in their assigned environments, we cannot exclude that the pigs were infected just prior to purchase for this study, since both serologic tests used to screen the piglets require at least 2 weeks after exposure to become positive. Regardless, the granges from which we sourced the pigs also strictly corralled the animals, so *T. hydatigena* and *E. granulosis* eggs were nonetheless able to bypass corralling and cause infection.

The overall seroincidence of developing antibodies on LLGP-EITB was approximately double for the FR pigs compared to that of corralled pigs. However, the pattern was different when looking at seroincidence trends over time. In the first 18 weeks after pigs were introduced into the village, the seroconversion rate was relatively low and similar across all 3 groups. After this point, there was a sharp divergence with FR pigs being nearly 6 times as likely to convert at any given moment compared to corralled pigs. This could be due to changes in pig behavior as they matured, roaming further and possibly becoming more likely to encounter human feces or eggs in the environment. GPS tracking studies of pig movement in northern Peru found that home range increased steadily as pigs aged, peaking in the 7–9 month age range [26]. However, seasonal and local geographic factors were more influential in general for determining roaming range and frequency of interaction with known defecation sites [27]. In our study seasonality was also important, as higher seroincidence in the second and third round of pig introductions corresponded primarily with the dry season.

Pigs in both netted and standard corrals seroconverted to positive on the LLGP-EITB, even though no *T. solium* cysticercosis was found in these animals on necropsy. This finding could have occurred if seropositive pigs with very low burden of cysts were missed on necropsy. We also cannot rule out the possibility of pig exposure to *T. solium* eggs through soil, water, or feed, which may have resulted in transient antibody development without established infection [28]. However, cross-reactivity of the LLGP-EITB with another cestode seems a more likely explanation. The GP50 band of LLGP-EITB is known to cross-react with *T. hydatigena* [17]. Given that *T. hydatigena* cysts were found among corralled pigs at necropsy, the positive LLGP-EITB is likely due to exposure to this, or potentially other, cross-reactive parasites.

## 5. Conclusions

In conclusion, we found no evidence in this study to suggest that either dung beetles or flies play a major role in transmission of *T. solium* to pigs. While corralling prevented infection with cysticercosis, it did not prevent infection with other cestodes or the development of antibodies suggesting exposure to Taenia eggs. However, while corrals also reduced risk of infection with other cestodes, as well as development of antibodies suggesting exposure, this protection was not complete. Mechanisms of *T. solium* egg dispersal in endemic settings other than insects should be evaluated to better understand transmission dynamics.

## Figures and Tables

**Figure 1 pathogens-12-00597-f001:**
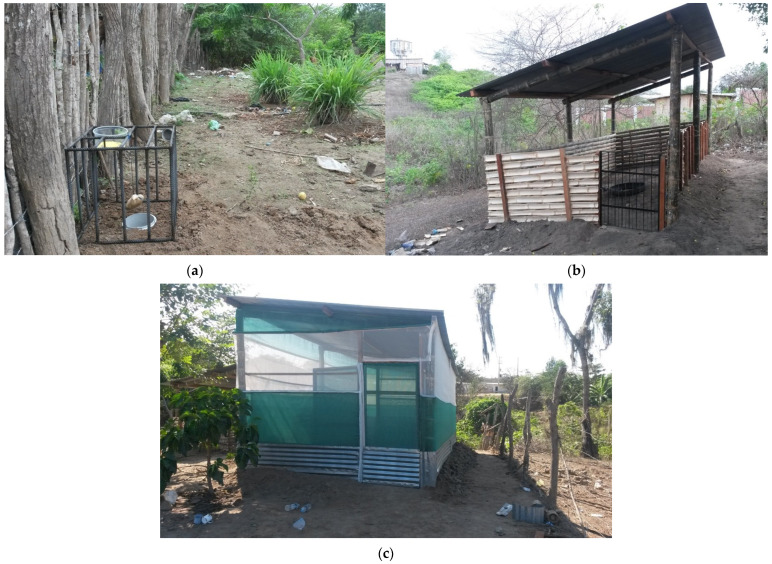
Representative photos of the three exposure environments. (**a**): Pigs in free-roaming environments were allowed to roam the village unrestrained. This image also shows placement of a typical fly and dung beetle trap. (**b**): Pigs in standard corrals were restricted to the corral space but were otherwise exposed to the open air. (**c**): Pigs in netted corrals were restricted to the corral space and were also protected from external flying and crawling insects. with double door to control the entrance of insects. Netted corrals were completely enclosed with dual layer polypropylene mesh netting, incorporated a separate double-door entry area, and staff were required to follow strict entry procedures including changing boots, examining the interior of the entry room for insects, and killing any insects found with insecticide spray before entering the protected corral space.

**Figure 2 pathogens-12-00597-f002:**
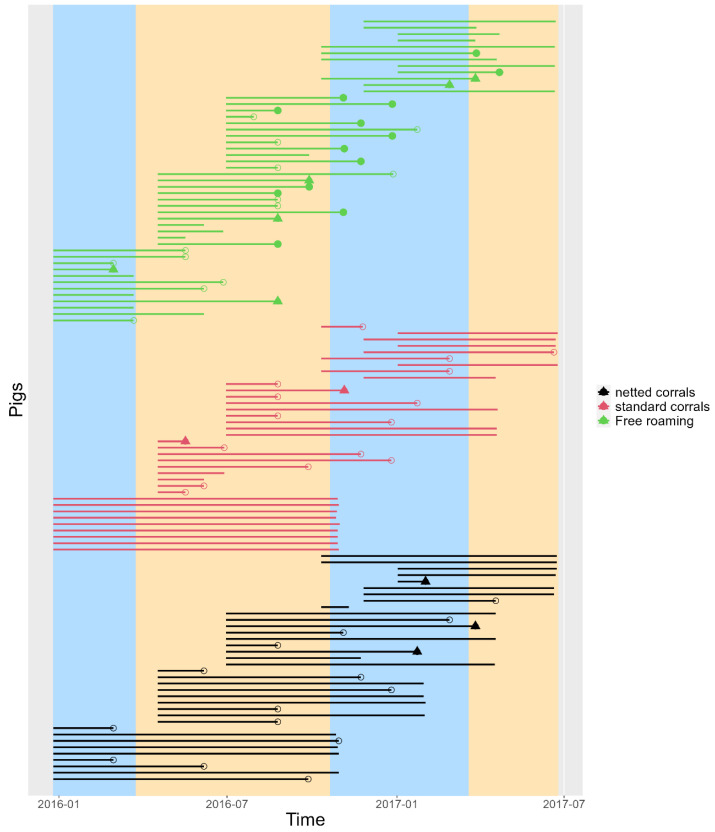
Individual survival time for each pig evaluated in the three cohorts. Vertical segments cover dry (**yellow**) and rainy (**blue**) seasons. The symbols at the end of each line are pigs that seroconverted (positive to LLGP-EITB), empty circles only seroconverted, filled circle also had cysts at necropsy and triangles had other cestode at necropsy. Lines without a shape at the end contributed to the model as right censored.

**Figure 3 pathogens-12-00597-f003:**
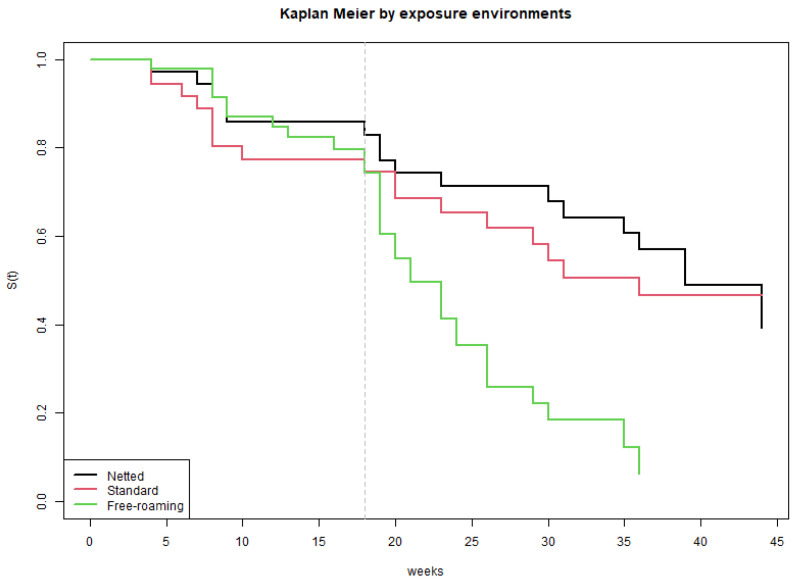
Kaplan–Meier survival curves representing time to development of a positive result on EITB among pigs in the three exposure environments. The dashed line at 18 weeks highlights the point at which the hazard for pigs in the Free Roaming environment increases and diverges with respect hazard in the other groups.

**Table 1 pathogens-12-00597-t001:** Total and EITB positive pigs, and *p*-value of log rank test stratified by exposure environment.

Variable	Categories	# of Pigs	EITB Positive (Failure)	Median Time (Weeks)	Log Rank Test (*p*-Value) *
Exposure environment	Netted corral	36	17	39	0.0005
	Standard corral	36	17	36	
	Free roaming	48	32	21	
Round	First	30	13	NC	0.008
	Second	30	21	23	
	Third	30	22	26	
	Fourth	30	10	NC	
Sex	Male	68	34	35	0.1
	Female	52	32	23	

* As shown by the results of the log rank test, the median time to positivity was statistically different for the variables ‘exposure environment’, ‘round’ and ‘sex’. NC = not calculated. # = Number.

**Table 2 pathogens-12-00597-t002:** Extension of the Cox proportional hazard (PH) for time-dependent variables model with three independent variables: breeding system, rounds, sex and time divided into two: before and after 18 weeks. Goodness of fit in Appendix A.

Variable	PH *	Lower 0.95	Upper 0.95	*p*-Value
**Round**				
Second/First	2.4214	1.2886	4.5499	0.006
Third/First	2.2624	1.1727	4.3649	0.015
Fourth/First	0.856	0.3497	2.0954	0.7
**Sex**				
Male/Female	0.6216	0.3889	0.9936	0.047
**Groups**				
Free roaming/Corrals (before 18 weeks)	1.1495	0.5254	2.515	0.7
Free roaming/Corrals (after 18 weeks)	5.6842	2.9237	11.0513	<0.0001

* PH = proportional hazard.

## Data Availability

The data collected for this study are available from the corresponding author upon request.

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
