# Peer review of "Evaluating the Role of Corrals and Insects in the Transmission of Porcine Cysticercosis: A Cohort Study"

_pathogens, 2023, doi:10.3390/pathogens12040597_

Round 1

Reviewer 1 Report

General:

This is a well-written article with a really thorough analysis of the data evaluating the role of insects and corrals in the transmission of T. solium to pigs that deserves publication. Surprisingly, no insects carrying Taenia eggs were found.

Specific comments:

Line 41: Specify some of those significant gaps.

Lines 98-99: Indicate the approximate prevalence of taeniasis in the 2 communities studied. These data will be important to understand subsequent results obtained for porcine cysticercosis.

Lines 161-162: Considering that some pigs became infected by other tapeworms, were they removed from the corrals for blood collection? The question posed is intended to assess that the pigs had no chance of becoming infected outside the corrals.

Lines 182-183: Liver is one of the earliest sites of Taenia infection in pigs, why were not livers studied?

Lines 335-336: According to the authors “The objective of this study was to evaluate the extent to which dung beetles and flies play a role in transmission of T. solium to pigs”. To analyze this epidemiological fact would only have required trapping insects, without the need to include pigs in corrals in the study, since if positive pigs and insects had been found, the route of infection of these pigs could have been through those insects, but also through the ingestion of eggs from soil, food, or water. Therefore, the sentence should be rephrased.

Lines 337-340: What other pathways are the authors referring to besides the presence of proglottids/eggs in human feces, and eggs in soil, food, water or dirty hands of the owner? Low-cyst burdens in infected pigs are precisely quite frequent in endemic areas due to the presence of non-infectious old eggs in the environment which once ingested by pigs act as a kind of vaccine causing this low number of cysticerci in pigs. Please, rephrase the sentence.

Lines 411-412: The fact that “Strict corralling prevented infection with cysticercosis in all pigs that were assigned to corrals, netted or not.” is obvious and completely expected and required no further verification. Therefore, rephrase the sentence along these lines as well.

Author Response

Dear Reviewer 1,

We thank you for your comments and suggestions that have contributed to improve the manuscript. We have modified the manuscript accordingly. All comments have been answered in this section and the modified sentences have been market in red.

Line 41: Specify some of those significant gaps.

Thank you for your comment. This information is detailed in the subsequent paragraph in the introduction, so we included a reference to this in parentheses. 

“…there are still significant knowledge gaps regarding how T. solium is transmitted between human and pig hosts at the community level These gaps (described below) limit the ability…”

Lines 98-99: Indicate the approximate prevalence of taeniasis in the 2 communities studied. These data will be important to understand subsequent results obtained for porcine cysticercosis.

The prevalence found in these villages was included.

“The study occurred in two adjacent rural villages from the Matapalo District in the Tumbes Region of northern Peru where T. solium is endemic (taeniasis prevalence 1.4-2.7%; unpublished).”

Lines 161-162: Considering that some pigs became infected by other tapeworms, were they removed from the corrals for blood collection? The question posed is intended to assess that the pigs had no chance of becoming infected outside the corrals.

This is an insightful question. All pigs were bled inside their respective corrals. We have detailed this in the manuscript as:

We collected a venous blood sample from the anterior vena cava of all cohort pigs each month, keeping the pigs inside their corrals to avoid the possibility of outside exposure or infection. All blood samples collected in the study were maintained temporarily on ice in coolers while teams were in the field,….”

Lines 182-183: Liver is one of the earliest sites of Taenia infection in pigs, why were not livers studied?

The livers were also dissected. All cysts found in the liver were collected, evaluated by PCR and sequenced. Although some liver lesions were positive to cox1 gen, none of them were confirmed as T. solium by sequencing. For that reason, we had not mentioned livers in the manuscript. In order to clarify that livers were indeed evaluated, we have adjusted the materials and methods section as follows:

“We conducted a detailed necropsy by systematically dissecting the entire carcass, and identifying any viable or degenerating cysts present in the brain, heart, liver, tongue and all skeletal muscles. Fine slices of less than 0.5 centimeters were used to dissect all inspected tissues.”

Lines 335-336: According to the authors “The objective of this study was to evaluate the extent to which dung beetles and flies play a role in transmission of T. solium to pigs”. To analyze this epidemiological fact would only have required trapping insects, without the need to include pigs in corrals in the study, since if positive pigs and insects had been found, the route of infection of these pigs could have been through those insects, but also through the ingestion of eggs from soil, food, or water. Therefore, the sentence should be rephrased.

“The objective of this study was to evaluate the extent to which dung beetles and flies play a role in transmission of T. solium to pigs.”

We do not agree with this comment and have chosen to keep the text unchanged. The presence of eggs in/on trapped insects alone would not indicate whether the eggs are infective, and whether this is a competent vector system.  For that reason, we considered it necessary to monitor pigs in standard corrals that are potentially exposed to insects harboring eggs, and to compare rates of seropositivity and infection among these compared to pigs in the other environments.

Lines 337-340: What other pathways are the authors referring to besides the presence of proglottids/eggs in human feces, and eggs in soil, food, water or dirty hands of the owner? Low-cyst burdens in infected pigs are precisely quite frequent in endemic areas due to the presence of non-infectious old eggs in the environment which once ingested by pigs act as a kind of vaccine causing this low number of cysticerci in pigs. Please, rephrase the sentence.

“On the contrary, the absence of pig cysticercosis infection in the standard corral environment, along with the complete absence of T. solium eggs in dissected beetles and of T. solium DNA in flies, suggest that other factors (eg. alternate routes of low-burden egg exposure, variability in egg infectivity, variability in pig susceptibility) are more likely to explain dispersion and low-cyst burdens of infected pigs.

Thank you for that comment. This point of view is interesting, and we now cite it as one of the possible contributing factors to low-cyst burdens, alongside other factors. Please note that, while the development of pig immunity through contact with old, non-infective eggs can be a reason, it is likely not the only reason of low cysts burden. As Ted Nash told me in the oral presentation of an earlier dose response paper: “experimental infection with a whole proglottid can cause less than ten cysts”. Hence, the open question may be: Is low burden primarily caused by dispersion or by variability in tapeworms?

Lines 411-412: The fact that “Strict corralling prevented infection with cysticercosis in all pigs that were assigned to corrals, netted or not.” is obvious and completely expected and required no further verification. Therefore, rephrase the sentence along these lines as well.

We have modified the sentence as suggested.

“In conclusion, we found no evidence in this study to suggest that either dung beetles or flies play a major role in transmission of T. solium to pigs. While corralling prevented infection with cysticercosis, it did not prevent infection with other cestodes or the development of antibodies suggesting exposure to Taenia eggs.”

Reviewer 2 Report

General comments:

Objective of the study is to evaluate the risk of porcine cysticercosis associated with exposure to human feces, dung beetles, and flies in an endemic community setting. Authors used a cluster-randomized cohort design to compare the risk of developing antibodies and infection among 120 piglets raised in either free-roaming, standard corral, or netted corral environments.

Researchers found no evidence in the study to suggest that either dung beetles or flies play a major role in transmission of T. solium to pigs. Researchers recommend that mechanisms of T. solium egg dispersal in endemic settings other than insects should be evaluated to better understand transmission dynamics.

Manuscript is well designed and written. The findings of the study are also very valuable in terms of evaluating the transmission routes of the disease.

Author Response

Dear Reviewer 2,

We appreciate your comments and suggestions, and we thank you for your time invested in reviewing the manuscript.